# Different Roles of Apoptosis and Autophagy in the Development of Human Colorectal Cancer

**DOI:** 10.3390/ijms241210201

**Published:** 2023-06-15

**Authors:** Giulia Orlandi, Luca Roncucci, Gianluca Carnevale, Paola Sena

**Affiliations:** 1Department of Surgery, Medicine, Dentistry and Morphological Sciences with Interest in Transplant, Oncology and Regenerative Medicine, University of Modena and Reggio Emilia, Via del Pozzo, 71-41124 Modena, Italy; giulia.orlandi@unimore.it (G.O.); gianluca.carnevale@unimore.it (G.C.); 2Department of Medical and Surgical Sciences, University of Modena and Reggio Emilia, Via del Pozzo, 71-41124 Modena, Italy; luca.roncucci@unimore.it

**Keywords:** human colorectal cancer, apoptosis, autophagy, oxidative stress, non-coding-RNA, p53, NF-kB, SQSTM1/p62

## Abstract

Colorectal cancer (CRC) remains a major life-threatening malignancy, despite numerous therapeutic and screening attempts. Apoptosis and autophagy are two processes that share common signaling pathways, are linked by functional relationships and have similar protein components. During the development of cancer, the two processes can trigger simultaneously in the same cell, causing, in some cases, an inhibition of autophagy by apoptosis or apoptosis by autophagy. Malignant cells that have accumulated genetic alterations can take advantage of any alterations in the apoptotic process and as a result, progress easily in the cancerous transformation. Autophagy often plays a suppressive role during the initial stages of carcinogenicity, while in the later stages of cancer development it can play a promoting role. It is extremely important to determine the regulation of this duality of autophagy in the development of CRC and to identify the molecules involved, as well as the signals and the mechanisms behind it. All the reported experimental results indicate that, while the antagonistic effects of autophagy and apoptosis occur in an adverse environment characterized by deprivation of oxygen and nutrients, leading to the formation and development of CRC, the effects of promotion and collaboration usually involve an auxiliary role of autophagy compared to apoptosis. In this review, we elucidate the different roles of autophagy and apoptosis in human CRC development.

## 1. Introduction

CRC remains a major life-threatening malignancy, despite numerous therapeutic and screening attempts [1]. During CRC development, programmed cell death plays a crucial role in the transformation process of benign polyps into adenomas and ultimately adenocarcinoma [2]. During the multistep process of human colorectal carcinogenesis, the key proteins regulating the programmed death of mucosal cells undergo fundamental expression changes. The cells of the normal mucosa evolve towards a neoplastic condition and acquire new characteristics and biological capabilities, among which the ability to escape cell death is one of the most important [3]. The multistep process of human carcinogenesis requires that cancer cells acquire the features enabling them to become tumorigenic and ultimately malignant. The apoptotic machinery is very complex and diversified and is composed of both upstream regulators and downstream effector components [4]. The existing balance between pro and anti-apoptotic factors belonging to the B-cell lymphoma-2 (Bcl-2 family) of regulatory proteins is the basis of the control of the “apoptotic trigger” that conveys signals between regulators and effectors. Moreover, several abnormality sensors that play key roles in tumor development have been identified, among which a DNA damage sensor that functions via the tumor suppressor p53 (TP53) is most notable [5]. Apoptosis and autophagy are two processes that share common signaling pathways, are linked by functional relationships and have similar protein components. During the development of cancer, the two processes can trigger simultaneously in the same cell, causing, in some cases, an inhibition of autophagy by apoptosis or apoptosis by autophagy [6]. In particular, several genes and proteins are part of the complex mechanism of autophagy. Recent evidence has demonstrated that autophagy plays a decisive role in all stages of human tumorigenesis, by leading initiation and contributing to progression and metastatic ability in tumors. The signal pathways that are activated in the autophagic process during carcinogenesis are not yet defined and appear to be dependent on the tumor microenvironment, as they can have both promotion and inhibition roles. Autophagy often plays a suppressive role during the initial stages of carcinogenicity, while in the later stages of cancer development, it can play a promoting role [6]. It is extremely important to determine the regulation of this duality of autophagy in the development of CRC and to identify the molecules involved, the signals and the mechanisms that allow autophagy to play a pro-malignant dominant role in some conditions and the opposite role in others. The most interesting research on autophagy in colorectal carcinogenesis has been focused on several molecules, mainly microtubule-associated protein 1A/1B-light chain 3 (LC3), sequestosome 1 (SQSTM1/p62) and nuclear factor-κB (NF-κB), and these studies have produced conflicting results. In this review, we try to elucidate the different roles of autophagy and apoptosis in human CRC development.

## 2. Apoptosis and Autophagy in Adenoma–Carcinoma Sequence

The crypts of the human colorectal mucosa exhibit a polarized structural organization. The reserve pool of the stem cells of the regenerative epithelium is very small and located at the base of the crypt. Stem cells move from the bottom of the crypt up, and on the apical margin of the mucosa about 10 million cells per day die from apoptosis, after which they fall into the lumen [7]. For this reason, adequate cell death is required to maintain the homeostasis of normal colorectal mucosa. However, defective signaling or unbalanced regulation of apoptosis is probably one of the causes of the onset and progression of an adenoma to the carcinoma sequence that results in CRC. It should be noted that the proteins most involved in apoptosis, such as Bcl-2 antagonist killer 1 (Bak) or Bcl-2, are not equally expressed in all anatomical regions of the colorectal mucosa, confirming the fact that there is a diversified regulation of death in the bowel [8,9]. Moreover, not only apoptosis, as a classical form of programmed cell death, has been shown to be operative and consistent in the colorectal glands, but also autophagy, a controlled process of cell self-digestion of great importance in situations of cellular stress, lack of nutrients, or energy deprivation (Figure 1) [10].

In healthy differentiated adult cells, autophagy delivers to the lysosome damaged cytosolic components and organelles, preventing ageing, and actively participates in intracellular quality control and renovation in order to avoid the formation of protein aggregates that could cause several diseases, including neurodegenerative ones [11]. In contrast to apoptosis, the intensity of autophagy cell expression decreases in the crypt from the bottom to the top. Autophagy is therefore confined to the lower part of the crypt [6]. Groulx et al. [12] suggested that, in the human colon, autophagy was physiologically present in the normal mucosa and is associated with a region containing the proliferative progenitor/stem cell populations, which are crucial for general metabolism and cellular turnover of the colonic epithelium. The integrity of the complex interaction between cell death and autophagy, a stress-reactive process, is necessary for the normal development and homeostasis of colorectal mucosa. Modifying the classic pathways of the cell death signal is often the trigger for the pathogenesis of various colorectal diseases, for example, chronic diseases and those sustained by a constant inflammation of the colorectal mucosa (Crohn’s disease as well as ulcerative colitis) [10]. Cancer cells evolve a variety of strategies to escape or circumvent the actions of the apoptotic machinery and program; moreover, autophagy appears to represent an additional barrier that must be overcome during cancer development and progression [13]. The development of CRC commonly requires three precisely connected steps: initiation, a process that modifies the molecular structure and phenotype of the normal cell; promotion, in which specific signal transduction cascades are activated; and progression, involving the activation of cells that have accumulated relevant gene mutations and phenotypic alterations (Figure 2). Changes in the expression pattern levels of different apoptotic proteins during the adenoma–carcinoma sequence highlights the crucial role of apoptosis during the progression of colon cancer. Moreover, both pro- and anti-apoptotic factors are upregulated in adenomas; therefore, tumor progression is probably due to a balance between these factors [14].

The tumor suppressor p53 function is frequently impaired in human cancers, and it is thought that p53 mutations play a pivotal role in the adenoma–carcinoma transition [15,16] and occur in 60% of reported CRCs [17]. For this reason, the transition from adenoma to carcinoma in the colorectal region is considered a mechanism in which the regulation of the apoptotic process, given its relationship to the p53 mutations, is essential [18]. As regards the autophagic process, its involvement in the initiation of the adenoma–carcinoma transition is still highly controversial. Studies of the relationship between autophagy and colorectal carcinogenesis have produced contradictory results: the conditional deletion of autophagy-related gene-7 (ATG7) in the intestinal epithelium does not increase the number of intestinal polyps [19], while the heterozygous deletion of autophagy-related gene-5 (ATG5)—an essential ATG protein in autophagy—in adenomatous polyposis coli (Apc) Min/+mice increases the number and size of adenomas through the activation of the epidermal growth factor receptor (EGFR)/extracellular signal-regulated kinase 1/2 (ERK1/2) and wingless and INT-1 (Wnt)/β-catenin pathways [20]. Li et al. [21] suggest that atractylenolide I, a natural substance that significantly decreased the viability of carcinoma cell lines through inducing intrinsic apoptosis, reduces intestinal adenoma formation by activating autophagy machinery to downregulate D-dopachrome tautomerase (D-DT). D-DT is a novel hypoxia-inducible gene and a direct target of hypoxia-inducible factor-1 alpha (HIF-1α) and hypoxia-inducible factor-2 alpha (HIF-2α) [22], and is highly expressed in two human CRC cell lines [21]. Studies, regarding the possible role of autophagy in the adenoma–carcinoma sequence initiation, indicate a possible activation of the immune system [23].

## 3. Regulatory Role of p53 in Apoptotic and Autophagic Processes

Over the years, the interest of researchers in determining different types of cell death has never diminished, since it remains an essential physiological process required to maintain tissue homeostasis. The recent recommendations of the Nomenclature Committee on Cell Death define autophagy-dependent cell death (ADCD) as regulated cell death that results from the autophagy machinery (i.e., pharmacological or genetic manipulations of autophagy genes block cell death), without involving alternative death pathways [24]. However, research attempts have not yet produced an accurate picture of cell death phenomena. The fate of a cell, whether it lives or dies, is crucial and can be the determining factor in the progression of tumors [25]. Incorrect regulation of the cell death signaling cascades is considered fundamental for the initiation and progression of CRC. Many in vitro studies have elucidated that the interplay between apoptosis and autophagy is highly complex and their influence in cancer can be modified by several factors, such as the types of tissue and cells involved, tumor stage and malignancy grading, type of oncogenic mutation, the entity of damage or cellular stress, intra-tumoral oxygen levels or the presence of nutrient deprivation [26]. The tumor suppressor protein, p53, presents a plethora of anticancer functions [27]. p53 is activated in the presence of cellular stress, including DNA damage, hypoxia, senescence and DNA repair, inducing cell cycle arrest to allow cell repair or apoptosis (mainly via the intrinsic pathway) [28]. The p53 gene is mutated in a high percentage of colorectal tumors and can be found in both adenomas and malignant cells, resulting in the disruption of tumor suppressor functions [24]. There is evidence that p53 mRNA expression may represent a useful survival predictor in patients with stage III CRC or rectal carcinoma [29]. On the other hand, Sakanashi F in one morphological study showed that autophagy and apoptosis are overrepresented in CRC and that p53 mutation may lead to the upregulation of autophagy [30]. p53 mutation may alter the ability to reduce autophagy, which implies that the p53 mutant increases autophagy, leading to more aggressive CRCs. Recently, researchers demonstrated that, after pharmacological or genetic protein kinase B (AKT) inhibition, p53 interacted with sirtuin 6 (SIRT6) and poly (ADP-ribose) polymerase1 (PARP1) directly to activate it, and promoted the formation of PAR polymer in human colon cancer cell lines. Consequently, the PAR polymer, after being transported to the outer membrane of the mitochondria, resulted in the release and transfer of apoptosis-inducing factor (AIF) into the nucleus and the promotion of cell death (Figure 3). These processes were inhibited by p53 deletion or mutation and protective autophagy was increased by the inhibition of AKT [31].

Another interesting paper showed that apoptotic cell death induced by N1, N11-diethylnorspermine (DENSpm) is boosted when autophagy is inhibited by 3-mase9pethyladenine (3-MA), an autophagy inhibitor, or beclin-1 siRNA through polyamine depletion or polyamine catabolic activation in colon cancer cells, regardless p53 mutation status. In conclusion, the researchers stated that p53 deficiency does not alter the response of colon cancer cells to apoptotic death induced by DENSpm in autophagy suppression conditions [32]. p53 is linked to the regulation of autophagy by affecting the availability of nutrients. The loss of p53 leads to the overproduction of LC3, a standard marker for autophagosomes, and forces cells to maintain elevated levels of autophagy. A study including HCT116 colorectal cells highlighted that p53 promotes selective downregulation of LC3 mRNA and protein in conditions of protracted nutrient deprivation. Initially, autophagy has a protective effect against cellular damage, avoiding nutrient starvation through the recycling of cellular substrate as proteins, protein complexes, lipids, ions and whole organelles. However, after complete nutrient depletion, this form of self-cannibalism is stalled, leading to the accumulation of aberrant metabolic intermediates that activate signaling promoting the execution of programmed cell death, such as apoptosis [33].

## 4. SQSTM1/p62 Is Related to Autophagy and Apoptosis in CRC Cells

SQSTM1/p62 is an autophagy receptor involved in optimizing the autophagosome formation process. After the fusion of autophagosomes with lysosomes, the autophagosome content, as well as SQSTM1/p62, is degraded [34]. SQSTM1/p62 plays specific and indispensable roles in selective autophagy and SQSTM1/p62 protein levels have been used as an indicator of autophagic flux [35]. The pivotal roles of SQSTM1/p62 in ingested protein catabolism through autophagy and lysosomal targeting have been well elucidated in the literature [34,35,36]. Although the role of SQSTM1/p62 in solid tumors is still debated, evidence has shown that SQSTM1/p62 is upregulated in different cancers and promotes tumor growth. Moreover, it has been assessed that SQSTM1/p62, as a tumor oncogene, is frequently abnormally upregulated and engaged in the acquired malignancy of gastrointestinal tumors, such as CRC [37]. However, the molecular mechanisms that regulate SQSTM1/p62 expression in CRC are still widely debated. Recently, Mukherjee et al. stated that, in HCT116 colorectal cells, SQSTM1/p62 is a mutp53 interactor, which associates selectively with the DNA contact mutant p53R273H but not with the structural mutant p53R175H. They further assess that the interaction with SQSTM1/p62 is needed for the ability of p53R273H to trigger cancer cell migration and invasion. The authors assumed that this acquired ability is due to the involvement of the mutp53-p62 axis in directing the ubiquitin-dependent proteasome degradation of connexin 43 and cell junction proteins [38]. Nevertheless, the autophagic process can be repressed by SQSTM1/p62 inhibition, leading to the arrest of cancer cell growth and the inhibition of cancer development [39]. On the other hand, literature has reported that SQSTM1/p62 expression in CRC cells was suppressed by the β-catenin/transcription factor (TCF)4 complex, which blocked phagocytic ingestion, but under stress conditions induced by nutrient depletion, SQSTM1/p62 was markedly raised, as β-catenin was immediately degraded by LC3 binding to the LC3-interacting region (LIR) of β-catenin [39]. Another interesting paper showed that poly C binding protein 1 (PCBP1), a novel tumor suppressor gene, withdraws or balances the basal cell autophagy necessary to retain cell homeostasis in CRC cells [40]. The authors stated that PCBP1 can affect various autophagy-related genes expression at the different autophagic stages, including the autophagosome formation, LC3B, the autophagosome-lysosome fusion and SQSTM1/p62. Moreover, the PCBP1–SQSTM1/p62 signaling axis may play a repressive role in the early phases of tumorigenesis. PCBP-1 acts by inhibiting both autophagy and proteasome-mediated SQSTM1/p62 degradation and stabilizing SQSTM1/p62 m-RNA. Moreover, considering that SQSTM1/p62 associates with caspase-8 and the subsequent apoptotic pathway [41], PCBP1 downregulation or depletion would be a relevant strategy for tumor cell survival via downregulation of SQSTM1/p62 and caspase-8, suggesting the concept that autophagy inhibition could be an efficient strategy to activate the apoptotic cell death mechanism in colorectal tumor cells [40]. In contrast, Zhang et al. [42] demonstrated that SQSTM1/p62 acts as an oncogene in CRC, both in vivo and in vitro, and that the overexpression of SQSTM1/p62 significantly inhibited apoptosis. The authors demonstrated that SQSTM1/p62 binds the vitamin D receptor (VDR) and may target the nuclear factor-erythroid-2-related factor 2 (NRF2)-quinone oxidoreductase-1 (NQO1) axis via VDR in human CRC cells, promoting CRC cell proliferation and invasion in vivo. Therefore, SQSTM1/p62 may act as a protective agent in tumor cell viability by triggering autophagy and blocking apoptosis under drug-induced stress. Interestingly, escin, a triterpene saponin, inhibited cell viability and colony formation in CRC cells. The treatment of escin induced DNA damage, leading to phospho ataxia-telangiectasia mutated (p-ATM) and histone γh2ax upregulation. In particular, the escin process increased the expression of p62, which had a protective effect against DNA escin-induced damage: knockdown of P62 increased DNA escin-induced damage, while overexpression of p62 decreased the rate of apoptosis, inducing DNA damage repair in escin-treated cells. Moreover, treatment with escin induced apoptosis in a manner directly proportional to concentration and time. Similarly, knockdown of SQSTM1/p62 produced an evident anti-cancer effect in vivo [43]. This result can be reported at the genetic level, since CRC cells treated with escin activated autophagy and overexpressed SQSTM1/p62 played an important protective role against escin-induced apoptosis and DNA damage, as SQSTM1/p62 suppresses the ataxia-telangiectasia mutated/phosphorylated histone family member X pathway [44]. Finally, SQSTM1/p62 may act as a shielded factor in cancer cell survival by triggering autophagy and blocking apoptosis under drug-induced stress; otherwise, it may act as a driving force to determine the adverse fate of tumor cells via mechanisms that remain partially unclear.

## 5. NF-κB as a Potential Matchmaker between Autophagy and Apoptosis

Studies on CRC have reported that the NF-kB signaling pathway promotes tumor initiation and contributes to cancer cell metastasis formation and epithelial to mesenchymal transition [45,46,47]. In the literature, there is a plethora of papers that confirm the role of NF-kB in regulating the apoptotic process. In fact, the molecular mechanisms by which NF-kB exerts its regulatory action are well known, for instance: NF-κB signaling prevents apoptosis by up-regulating anti-apoptotic genes expression such as B-cell lymphoma-extra-large (Bcl-xL), the Bcl-2-related gene (A1/BFL1), cellular inhibitors of apoptosis proteins (cIAPs) and caspase-8/FAS-associated death domain-like IL-1beta-converting enzyme inhibitory protein (c-FLIP) [46]. Over the past few years, some very interesting studies have highlighted a correlation between NF-kB, autophagy and apoptotic processes. A study showed that metformin had an antiproliferative effect related to changes in the expression of nuclear factor E2-related factor (NRF-2)/NF-κB pathways on human colon cancer cells (HT-29) in a dose- and time-dependent manner, and exerted growth inhibitory effects by increasing both apoptosis and autophagy [48]. Recently, the autophagy-regulated cytotoxic effect of green synthesized silver nanoparticles (Brassica Ag-NPs) against human epithelial colorectal adenocarcinoma cells was demonstrated. The authors found that Brassica Ag-NPs induced NF-κB mediated autophagy in Caco-2 cells. The decreased expression of NFκB was associated with an increased expression of inhibitor of NF-κB (IκB)-kinase, which is involved in autophagic process initiation. Moreover, the activity of p53 and light chain 3 (LC3) II greatly accelerated autophagosome formation, and the inhibition of Akt and mammalian target of rapamycin (mTOR) was evident. In conclusion, the authors stated that, in the event of excessive activation of the autophagic process, after the complete depletion of nutrients, this process stalled, leading to the accumulation of aberrant metabolic intermediates that can trigger apoptotic cell death, which subsequently resulted in necrosis [49]. A study examining the relationship between long non-coding RNA cancer susceptibility candidate 2 (lncRNA CASC2) and microRNA19 (miR19) in colon cancer reported that lncRNA CASC2 suppressed expressions of LC3 and SQSTM1/p62 and significantly reduced cell viability of colon cancer cells, promoting apoptosis. Overexpressed miR19a reduced the expression of lncRNA CASC2 and increased cell viability. NF-kB was upregulated in colon cancer cell lines; otherwise, after treatment by N4-(4-phenoxyphenethyl) quinazoline-4,6-diamine (QNZ), a NF-kB inhibitor, the expression of Bcl2 was downregulated and Bax expression was upregulated. As for autophagy, expressions of LC3 and SQSTM1/p62 were reduced by inhibition of NF-kB. Therefore, inhibition of NF-kB reversed the role of lncRNA CASC2 and enhanced the effect of miR19a. The authors deduced that lncRNA CASC2 and miR19a might have a close correlation during the progression of colon cancer cells and that NF-kB was the signaling pathway, which could regulate the correlation between lncRNA CASC2 and miR19a [50]. Researchers discovered that bupivacaine, a local anesthetic, inhibited colorectal adenocarcinoma cell proliferation and migration, which are related to increased endoplasmic reticulum stress [51]. It seems that bupivacaine significantly inhibited CRC cell vitality, promoted the apoptosis rate and expression of apoptosis genes such as Bax and caspase-3, inhibited Bcl-2 expression, prevented cancer cell migration, increased autophagy-related protein LC3B II/LC3B I ratio and Beclin-1 expression and hindered p62 expression. Finally, bupivacaine could influence the expression of inflammatory factors such as interleukin 1 beta (IL-1 beta), interleukin 6 (IL-6) and tumor necrosis factor alpha (TNF-α), increasing them and inhibiting phosphorylation of proteins I kappa B kinase (IKK), IκB and NF-κB, related to the signaling pathway of NF-κB in SW480 and SW620 cells. In in vivo experiments, bupivacaine prevented tumor volume increase, as well as NF-κB expression [52].

## 6. Oxidative Stress and Modulation of Autophagy and Apoptosis

Oxidative stress has been implicated in the pathophysiology of cancer: high levels of intra-cytoplasmatic reactive oxygen species (ROS), generated by accelerate aerobic glycolysis followed by “selfish” metabolic reprogramming (the Warburg effect), increase oncogene activity, the activation of growth factor dependent pathways or the presence of an increase in the pool of oxidizing enzymes, leads to genetic instability [53,54]. Aerobic glycolysis or the Warburg effect is a hallmark of metabolic phenotypes of cancer; in fact, cancer cells, including CRC cells, present altered glucose metabolism and are characterized by an enhanced uptake of glucose and an increased conversion of glucose to lactate [55]. In the past few years, many studies have focused on the importance of the role played by ROS in the crosstalk between autophagy and apoptosis of colon cancer cells. ROS has a negative feedback action on the autophagic process: ROS can promote autophagy, and at the same time, autophagy reduces ROS production by removing damaged mitochondria, endoplasmic reticulum and other materials that contribute to ROS generation [56,57]. On the other hand, ROS can trigger DNA damage, initiating the apoptotic process pathway in colon cancer cells [58,59]. The most common method by which ROS delete transformed cells is the activation of programmed cell death, which is completed by an extrinsic or an intrinsic pathway; both pathways culminate in caspase-induced final cell demise with the formation of apoptotic bodies that are removed by adjacent phagocytes (Figure 4) [60].

In the past few years, several interesting articles have been published concerning the inducing effect on both apoptosis and autophagy of drugs, mostly of natural origin, through regulatory pathways mediated by oxidative stress [61,62,63,64,65]. In this regard, a study investigated how β-elemene, extracted from the genus Curcuma, affected cells in vitro and in vivo. The results reported that after β-elemene treatment, CRC cells exhibited apoptotic bodies and increased levels of cleaved-caspase-3/9 and PARP proteins. The in vivo experiments conducted on mouse-xenograft models inoculated with mouse colon carcinoma HT-29 cells presented the same expression pattern of apoptotic markers. Moreover, β-elemene promoted the formation of intracellular acidic vesicle organelles (autophagy lysosomes) and boosted the expression of autophagic markers such as LC3B and SQSTM1/p62 in human CRC cells. In addition, β-elemene increased the expression of LC3B-II in tumor tissues, indicating that the autophagosomes–lysosomes fusion is very active. Moreover, the authors stated that β-elemene increased ROS intracytoplasmic levels, enhanced the phosphorylation of activated AMP-activated protein kinase (AMPK) and reduced the phosphorylation of mTOR in human CRC cells; hence, β-elemene leads to autophagy and apoptosis via the activation of the ROS/AMPK/mTOR pathway in human CRC cells [66]. Meanwhile, other authors demonstrated that polysaccharide from Dendrobium officinale (DOP) indirectly targeted mitochondria and induced excess ROS production, which subsequently caused mitochondrial dysfunction and inhibited ATP synthesis. Then, the low ratio of ATP/AMP activated AMPK/mTOR signaling, which in turn increased autophagy in CT26 cells. Finally, excessive activation of the autophagic process induced cancer cell death [67]. Similarly, polyphyllin I (PPI) induces ROS production and down-regulates the AKT/mTOR pathway, which in turn triggers the autophagic cell death and apoptosis of SW480 colon cancer cells [68]. Intriguingly, other articles emphasize the protective role played by the autophagic process on the survival of colon cancer cells treated with anticancer drugs. In fact, Antrodia salmonea (AS), a traditional Chinese medicinal fungus with anti-cancer proprieties, reduced the phosphorylation of AKT and mTOR in SW620 cells, inhibiting AKT/mTOR signaling cascades. Moreover, AS decreased cell viability and induced apoptosis and cytoprotective autophagy via intra-cytoplasmatic ROS buildup and interruption of the signaling pathways for ERK [69]. In the literature, it was reported that selenium exhibited chemo-preventive and chemotherapeutic effects against CRCs [70]. As regards this topic, Yu et al. identified a signaling pathway involved in selenite-induced protective autophagy in CRC cells undergoing apoptosis. Selenite leads to ROS production and promotes phosphorylation of AMPK, which binds and activates forkhead box O3 (FoxO3a). The phosphorylated FoxO3a binds to γ-aminobutyric acid receptor-associated protein-like 1 (GABARAPL-1) promoter and upregulates the transcription level of GABARAPL-1, activating an autophagic process that serves as a survival signal against apoptosis in both CRC cells and colon xenograft models [71].

## 7. Role of Non-Coding RNA in Enhancing or Inhibiting Autophagy and Apoptosis

With the deepening of the knowledge about the interaction between autophagy and apoptosis in CRC, researchers gradually begin to pay attention to the role of non-coding-RNA in this interplay. microRNAs (miRNAs) are a group of evolutionarily conserved small RNAs, endogenous, non-coding, single-stranded RNAs that regulate the expression of their target genes through mRNA degradation and serve a negative role at post-transcriptional levels. miRNAs have been found to influence many cellular biological processes including apoptosis, cell cycle distribution and autophagy in CRC [72,73,74,75] (Table 1 and Table 2). In detail, as regards autophagy, miRNAs may play a dual, and therefore conflicting, role in tumor cell proliferation through modulation of the autophagic process. While miR-211 promote CRC proliferation by inhibiting autophagy and targeting tumor protein 53-induced nuclear protein 1 (TP53INP1) [76], on the other hand, miR-30d suppresses cell proliferation in CRC by inhibiting autophagy. The authors demonstrated that miR-30d inhibited autophagy and promoted apoptosis of colon cancer cells by directly targeting ATG5, phosphatidylinositol-4,5-bisphosphate 3-kinase catalytic subunit beta (PIK3CB) and beclin 1 (BECN1); the inhibition of proliferation was due to negative regulation of cell autophagy and promotion of cell death [77]. More recently, a study focused on the effect of miR-126 on the autophagy and apoptosis of CRC cells in vitro and in vivo highlighted the synergistic role of the two processes in the inhibition of CRC cell proliferation by modulating the mTOR pathway [78]. Otherwise, overexpressing miR-142-3p facilitated proliferation and inhibited apoptosis of CRC cells. miR-142-3p targeted and decreased TP53INP2 levels. On the other hand, TP53INP2 overexpression suppressed the HEDGEHOG signaling pathway and induced the activation of CRC cell autophagy. In conclusion, miR-142-3p hindered tumor cell autophagy and promoted CRC progression via targeting TP53INP2 [79]. Moreover, a very intriguing study of Zhang et al. has shown the interplay between long non-coding RNA growth arrest specific 5 (GAS5) and miR-34a, and their regulative role on macroautophagy and apoptosis in CRC patients [80]. GAS5 is a newly discovered lncRNA that exerts inhibitory effects through the modulation of several target microRNAs in cancer [81], and miR-34a acts as a tumor-suppressor gene in human CRC by blocking autophagy through sirtuin 1 (SIRT1) regulating [82]. Finally, this study demonstrated that miR-34a participates in regulating GAS5-suppressed CRC cell macroautophagy and leads to apoptosis through the mTOR/SIRT1 pathway. GAS5-mediated regulation of macroautophagy retains CRC cells in an equilibrium state that protects against apoptosis. Consistent with this notion, miR-483 promotes the development of CRC by inhibiting the expression level of etoposide-induced 2.4 (EI24), an autophagy associated transmembrane protein [83], and the knockdown of homeobox transcript antisense intergenic RNA (HOTAIR), a well-known lncRNA, stimulates cell apoptosis and suppresses cell autophagy by upregulating microRNA-93 and downregulating autophagy-related 12 (ATG12) in CRC [84]. Circular RNA (circRNA) is a kind of high abundance non-coding covalent closed RNA formed by exon sequence and intron sequence, and many studies have shown that a variety of circRNA is abnormally expressed in CRC, and its abnormal expression is related to the proliferation, apoptosis and autophagy of CRC cells [85]. Despite the relationship between circRNAs and apoptosis in CRC having been confirmed by some studies, their regulation of autophagy molecules is gradually being revealed. In fact, silencing of circular homeodomain-interacting protein kinase (circ HIPK3) and hsa_circ_0007534 induces apoptosis in HCT116 and HT-29 CRC cells [86,87]. In the past few years, many circular RNAs have been identified, the downregulation of which induces the activation of the apoptotic process in CRC cells, i.e., hsa_circ_0020397, hsa_circ_0000523 [88,89]. Recently, Chen et al. showed that silencing circular RNA ubiquitin-associated protein 2 (circUBAP2) decreased the formation of total autophagosomes and autolysosomes in CRC cells, and when circUBAP2 is knocked down, a decrement of LC3B-II expression, a downregulation of LC3B-I/II conversion and the degradation of autophagy-related protein 6 (beclin1), autophagy-related protein 7 (ATG7) and forkhead box O1 (FOXO1) proteins were observed. In conclusion, overexpression of circUBAP2 can effectively activate autophagy in CRC cells by targeting the miR-582-5p/FOXO1 axis [90]. Authors found that circCCDC66 in CRC under hypoxic conditions promoted the malignant behaviors of CRC cells by regulating the miR-3140/autophagy pathway [91]. Moreover, circular autophagy-related protein 4B (circATG4B) encodes a novel functional protein (circATG4B-222aa) that promoted autophagy in CRC cells both in vitro and in vivo. Therefore, circATG4B-222aa interacts competitively with transmembrane p24 trafficking protein 10 (TMED10) and functions as a decoy that prevents TMED10 from binding to ATG4B, leading to increased autophagy [92].

## 8. Pathways Implicated in Regulation of Apoptosis and Autophagy and Possible Therapeutic Targets

The most important pathway involved in the induction and regulation of apoptosis and autophagy in CRC is the Wnt pathway and its component components, such as β-catenin, disheveled (Dvl), adenomatous polyposis coli (APC) and axin, that are activated along with the Wnt signaling cascade during cancer development [93]. Wnt signal transduction is typically divided into classical and non-classical pathways. The classical pathway is involved in cell survival, proliferation, apoptosis and autophagy, while the non-classical pathway regulates cell polarity and migration. The Wnt/β-catenin signaling and autophagy pathways play important roles in tissue homeostasis and tumorigenesis. A variety of studies, reporting experimental design of the genomic manipulation of β-catenin expression levels in vitro and in vivo, show that β-catenin inhibits autophagosome formation and directly blocks SQSTM1/p62 via T cell-specific DNA-binding protein (TCF4) [94]. During starvation, β-catenin is selectively degraded through the formation of β-catenin-LC3 complexes, which avoid β-catenin/T cell-specific DNA-binding protein (TCF) mediated transcription and proliferation in order to better adapt to metabolic stress conditions. The formation of the β-catenin-LC3 complex is mediated by the W/YXXI/L motif and the LC3 interaction region (LIR) in β-catenin. Therefore, as previously described, Wnt/β-catenin inhibits autophagy and SQSTM1/p62 expression, while β-catenin itself is targeted for autophagic degradation [95]. Autophagy negatively controls Wnt signaling by promoting Dvl degradation. Von Hippel–Lindau protein-mediated ubiquitination is necessary for the binding of Dvl2 to SQSTM1/p62, leading to the formation of Dvl2 aggregates under conditions of nutrient deprivation and LC3-mediated autophagosome recruitment. Finally, the ubiquitylated Dvl2 aggregates are degraded through the autophagy-lysosome pathway [96]. Interestingly, an inverse correlation between Dvl expression and autophagy was found in late stages of CRC development, supporting the hypothesis that autophagy may contribute to the aberrant promotion of Wnt signaling during tumorigenesis [93]. In the literature, many reports indicate that Wnt/β-catenin signaling plays a pivotal role in tumor initiation and development, and its function of inhibiting cell apoptosis may result in tumor chemoresistance [97]. As regard autophagy, APC, a modulator of the Wnt signaling pathway, can interact with other members of the Wnt cascade to regulate the development of bowel diseases by affecting autophagy [98]. Recently, researchers showed that Na/H exchange 3 regulator 1 (NHERF1), which is normally lost or down-regulated in early dysplastic adenomas, triggers nuclear β-catenin activity. The depletion of NHERF1 is responsible for the exacerbation of the transformed phenotype in vitro and in vivo in CRC cells. The genetic or pharmacologic manipulation of β-catenin caused an autophagy-to-apoptosis switch’ that implied the activation of caspase-3, the cleavage of PARP and the decrement of phosphorylated ERK1/2, beclin-1 and ras-related in brain 7 (Rab7) autophagy proteins levels, which in turn strikingly impacted the fate of CRC cells. Finally, the researchers hypothesized that doubleβ-catenin/NHERF1-inhibitory strategy may be fruitful to augment the induction of apoptotic death in CRC cells refractory to Wnt/β-catenin-targeted therapeutics [99]. Casein kinase 1 α (CK1α) is an enzyme that simultaneously regulates both Wnt/β-catenin and AKT. Despite the link of the AKT and Wnt pathways to autophagy in RAS-mutated CRC cells not being well defined, a pharmacological study demonstrated that CK1α is involved in regulation of autophagy, Wnt/β-catenin and AKT pathways in rat sarcoma (RAS)-mutant CRC cell lines. In particular, CK1α inhibition significantly reduced the AKT/phospho-β-catenin axis, and at the same time blocked the autophagy flux. In addition, CK1α inhibition induces the activation of apoptotic machinery [100]. According to recently published literature, achaete-scute homolog 2 (Ascl2), as the target molecule of the Wnt signaling pathway, is an important marker of colon cancer stem/precursor cells [101]. Wang et al. [102] showed that the suppression of Ascl2 expression exerts a tumor suppressor function in CRC by inducing excessive autophagy. Apoptosis, which is triggered by si-Ascl2 (small/short interference), could be counteracted by treatment with autophagy inhibitors, such as 3-methyladenine (3-MA) and chloroquine (CQ), indicating that Ascl2 targeted therapy may represent a new strategy for the treatment of CRC. Studies published thus far established that SLC6A14, a Na^+^/Cl^−^ coupled neutral and cationic amino acid transporter, is up-regulated in CRC tissues and in colon cancer cell lines. Recently, researchers have paid attention to the evaluation of the impact of SLC6A14 deletion on colon cancer in multiple preclinical model systems, comparing the effects with pharmacologic SLC6A14 blockade. Treatment of colon cancer cells with α-methyltryptophan (α-MT), a blocker of SLC6A14, decreases mTOR activity, triggers autophagy and promotes apoptosis. Moreover, in xenograft and syngeneic mouse tumor models, silencing of SLC6A14 or suppressing its action by α-MT reduces the size of tumor via targeting of β-catenin [103]. Tumor necrosis factor-α-inducing protein 8-like 2 (TIPE2) is a novel potential therapeutic target for advanced and recurrent human rectal adenocarcinoma [104]. TIPE2 overexpression modulates apoptosis through the Wnt/β-Catenin signaling by down-regulating the expression levels of Wnt3a, phospho(p)-β-Catenin and p-glycogen synthase kinase-3β. Family with sequence similarity 134, member B (FAM134B), in CRC, act as a cancer suppressor and inhibits autophagy by regulating the autophagic turnover of endoplasmic reticulum [105]. A fairly complex experimental study has demonstrated that exogenous inhibition of FAM134B leads to upregulation of EB1, which could in turn promote the WNT/β-catenin pathway by inactivating the tumor suppressor gene APC and consequently activating β-catenin in colon cancer cells. FAM134B is probably involved in the activation and regulation of earlier events/steps in the adenoma–carcinoma sequence of colorectal tumorigenesis by affecting autophagy via the WNT/β-catenin pathway [106].

## 9. Discussion

It is necessary to revisit, update and understand the main pathways and components that control cell death and autophagy, the connection between the pathways associated with CRC and the possible interaction between these pathways to obtain a better understanding of the biology of cell death in CRC context. Apoptosis plays an important role in many biological events, including the elimination of harmful cells. Malignant cells that have accumulated genetic alterations can take advantage of any alterations in the apoptotic process and, as a result, progress easily in the cancerous transformation [6]. A dysregulation of autophagic flux leads to an intracytoplasmic accumulation of organelles, protein aggregates and metabolic intermediates. These accumulations may trigger the over-production of reactive oxygen species and cause metabolic insufficiency. Especially in stressful situations and in conditions of nutrient deprivation, impairment of autophagic flux can promote tumorigenesis [107]. By contrast, autophagy is essential for the survival of cancer cells and in malignant cells usually a high level of autophagy is evident. However, autophagy induction promotes cancer cells survival under conditions of hypoxia and metabolic and energy depletion. As regards CRC, autophagosome formation is most prominent in tumors cells growing in a hypoxic environment [108]. Importantly, there is a wide overlap of the apoptosis and autophagy signaling networks and providing mechanistic insight their relationships is very difficult. Most prominently, Bcl-2 proteins are at the crossroad between the two processes; they are inhibitors of both apoptosis and autophagy by binding pro-autophagic beclin1 [109]. Therefore, it has been shown that dual specificity protein phosphatase 4 (DUSP4) silencing blocked the protein interaction between Bcl-2 and beclin1 or Bax in HCT116 cells. Moreover, the survival of HCT116 cells inhibited by DUSP4 silencing was prevented by autophagy impairment with spautin-1. DUSP4 can promote the viability and function of CRC cells by inhibiting Bcl-2 phosphorylation-dependent autophagic cell death and apoptosis [110]. Colitis and colon cancer initiation are prevented by the activation of intestinal epithelial autophagy; however, colon cancer growth and progression are aided by this process [47]. The relationship between autophagy and apoptosis is complex in CRC because normally the two processes are active in the same cells, but not simultaneously: autophagy always precedes the initiation of apoptosis [9]. At the same time, a variety of signal transduction pathways and regulators are involved and autophagy may induce or inhibit apoptosis depending on the cell type involved and the nature and duration of the stimulus/stress [10,111,112]. For these reasons, depending on the type of interaction between autophagy and apoptosis, an antagonistic or synergistic effect may occur in CRC cells. As described previously, excessive autophagy induces cancer cell death through the PI3K/AKT/mTOR pathway, which plays different roles in autophagy and apoptosis of CRC [68]. Autophagy and apoptosis can comparably induce cancer cell death in CRC. Another instance of cooperation between the two processes is represented by the induction of apoptosis in HT-29 and Caco-2 human cells by Rhus coriaria extract (RCE). RCE significantly impaired the viability of colon cancer cells through the caspase-7-dependent pathway, but it does not represent the main mechanism of cell death, which is triggered through programmed cell death type II (PCD-II), probably due to excessive autophagy activation [113].

## 10. Conclusions

All the reported experimental results essentially indicate that, while the antagonistic effects of autophagy and apoptosis occur in an adverse environment characterized by deprivation of oxygen and nutrients leading to the formation and development of CRC, the effects of promotion and collaboration usually involve an auxiliary role of autophagy compared to apoptosis in colon cancer cells. In an overwhelming majority of cases, studies examining the mechanistic role of autophagy and apoptosis on CRC tumorigenesis used xenograft mice models and CRC cell lines. Moreover, many of these studies have targeted autophagy and apoptosis transiently through genetic manipulations or the use of drugs, which can influence the study in various ways. From our point of view, it would be necessary for preclinical research to focus on the improvement of gene-editing techniques, which are necessary to delineate the mechanistic role of autophagy and apoptosis in CRC. The results obtained at this stage would be of great use in clinical trials in patients to test the response to immune-checkpoint inhibitors. In addition, it is necessary to focus attention on ongoing clinical trials that are using apoptosis or autophagy inhibitors in combination with chemotherapeutics or drugs. These studies aim to capture the positive results of using autophagy as a process of inducing programmed cell death in cancer cells. Such autophagic manipulation can lead to promising effects in the treatment of CRC if it will attempt to solve the autophagy paradox that migrates from an anti-cancer to a pro-cancer mechanism, so treatments need to be highly specific for the given setting.

We believe that it will also be a major challenge to address the interaction pattern between autophagy and apoptosis, as well as the regulatory signaling network and cooperating factors involved, which could play a pivotal role in the management of CRC. In this perspective, autophagy and apoptosis can be used conveniently in the specific context of pharmacological interventions as new targets to provide a basis for the development of drugs in the field of CRC.

## Figures and Tables

**Figure 1 ijms-24-10201-f001:**
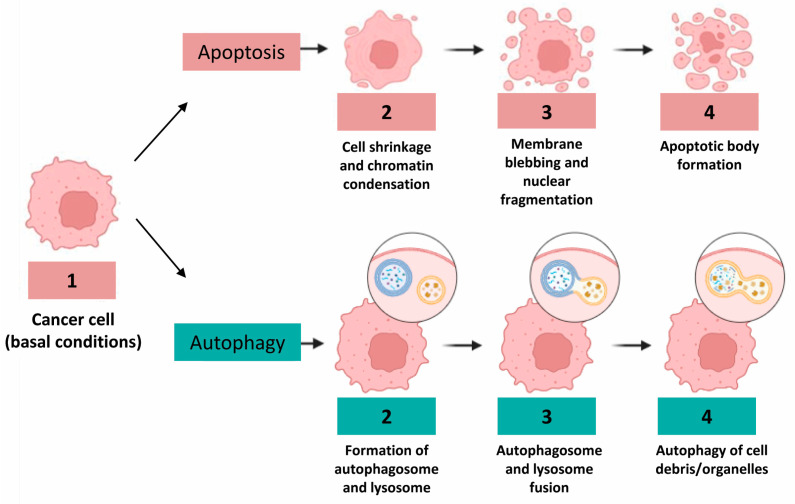
Different morphological features of human colon cancer cells during apoptosis and autophagy considered as prototypic cell death processes, in particular, morphogenetic changes occurring in the nucleus during apoptosis, that is, fragmentation process or, conversely, vacuole formation during autophagy.

**Figure 2 ijms-24-10201-f002:**
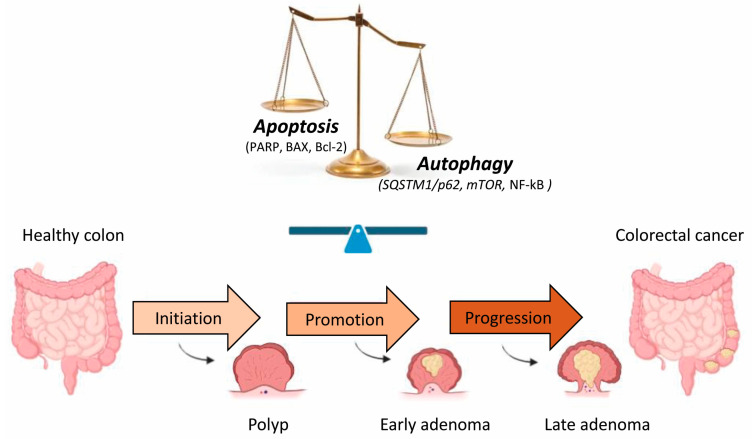
Schematic illustration of the human CRC sequence and development. There are three stages in the development of colorectal carcinogenesis: initiation, promotion and progression. Changes in the expression of different apoptotic and autophagic proteins during the transformation of adenoma into CRC suggest the importance of the balance between these two events in the progression of CRC. [BAX, Bcl-2 associated X protein; Bcl-2, B-cell lymphoma-2; *mTOR*, mammalian target of rapamycin; NF-κB, nuclear factor-κB; PARP, poly (ADP-ribose) polymerase, *SQSTM1/p62*, sequestosome 1].

**Figure 3 ijms-24-10201-f003:**
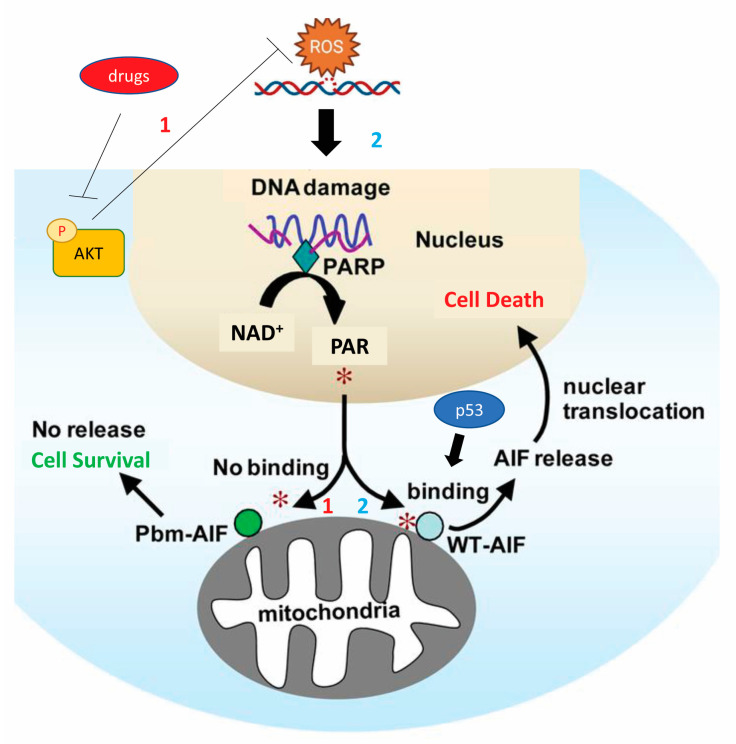
Mechanism by which p53 promotes the apoptosis of human colon cancer cell lines by releasing and transferring AIF into the nucleus after the formation of the PAR polymer. This death signaling pathway is inhibited by p53 deletion or mutation and protective autophagy is increased by inhibition of AKT. *1, No binding between PAR and AIF; *2, binding between PAR and AIF [AIF, apoptosis-inducing factor; AKT, protein kinase B; NAD, Nicotinamide adenine dinucleotide; p53, protein-53; PAR, poly (ADP-ribose); PARP, poly (ADP-ribose) polymerase].

**Figure 4 ijms-24-10201-f004:**
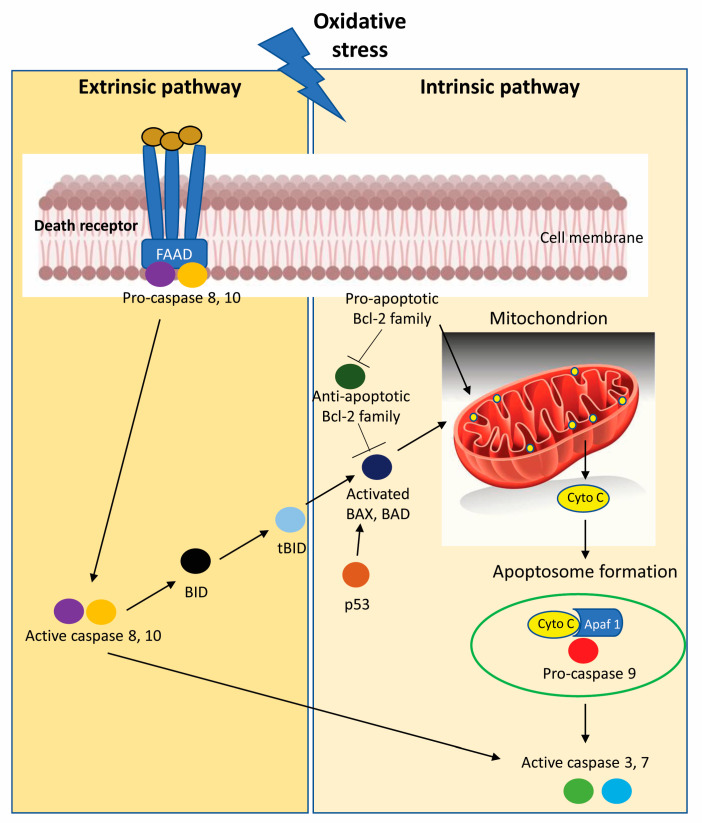
Oxidative stress promotes the activation of programmed cell death initiated by intrinsic apoptotic signaling in the mitochondria or extrinsic apoptotic signaling by death receptor pathways in CRC cells. [Apaf 1, Apoptotic protease activating factor 1; BAD, Bcl-2-Associated agonist of cell death; BAX, Bcl-2-associated X protein; Bcl-2, B-cell lymphoma-2; BID, BH3-interacting domain death agonist; cyto c, cytochrome c; tBID, C-terminal fragment termed truncated Bid].

**Table 1 ijms-24-10201-t001:** Non-coding-RNA associated with apoptosis in CRC.

Non-Coding-RNA	Regulation	Target	References
let-7e	down	DCLK1	[72]
miR-107	up	PAWR	[72]
miR-10b	up	KLF4	[72]
miR-135b	up	FOXO1	[72]
miR-15b-5p	down	XIAP	[72]
miR-184	down	MYC, BCL2	[72]
miR-198	down	ADAM28	[72]
miR-200b-3p	down	WNT1	[72]
miR-217	down	MAPK1	[72]
miR-217-5p	down	PRKCI, BAG3, ITGAV, MAPK1	[72]
miR-218	down	CFLAR	[72]
miR125b	up	STAT3	[73]
miR766-3p	up	TGFBI	[74]
miR1184	up	CSNK2A1	[75]
miR30d	up	ATG5, PIK3CB, BECN1	[77]
miR126	up	mTOR	[78]
miR142-3p	down	TP53INP2	[79]
lncRNAGAS5	down	mTOR/SIRT1	[80]
miR34a	up	SIRT1	[82]
lncRNAHOTAIR	down	ATG12	[84]
miR93	up	BCL2	[84]
CircHIPK3	down	miR-7	[86]
hsa_circ_0007534	down	BCL2	[87]
hsa_circ_0020397	down	PDL-1	[88]
hsa_circ_0000523	down	WNT/Bcatenin	[89]
circCCDC66	down	miR-3140	[91]

**Table 2 ijms-24-10201-t002:** Non-coding-RNA associated with autophagy in CRC.

Non-Coding-RNA	Regulation	Target	References
miR-125b	up	APC	[72]
miR-20a	down	ATG5, RB1CC1	[72]
miR-210	up	BCL2	[72]
miR-214	down	ATG12	[72]
miR-216a	down	MAP1S	[72]
miR-218	down	YEATS4	[72]
miR-22	down	BTG1	[72]
miR-30a	down	BECN1	[72]
miR-338-5p	up	PIK3C3	[72]
miR-409-3p	down	BECN1	[72]
miR-502	down	RAB1B	[72]
miR221	down	TP53INP1	[76]
miR30d	down	ATG5, PIK3CB, BECN1	[77]
miR126	up	mTOR	[78]
miR142-3p	down	TP53INP1	[79]
lncRNAGAS5	up	mTOR/SIRT1	[80]
miR34a	down	SIRT1	[82]
miR483	down	EI24	[83]
lncRNAHOTAIR	down	ATG12	[84]
miR93	up	BCL2	[84]
circUBAP2	up	LCB3II, FOXO1, BECLIN1	[90]
circATG4B	up	ATG4B	[92]

## Data Availability

No new data were created.

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
