# Peer review of "Different Roles of Apoptosis and Autophagy in the Development of Human Colorectal Cancer"

_ijms, 2023, doi:10.3390/ijms241210201_

Round 1
Reviewer 1 Report
Dear Authors,
The submitted manuscript discusses in detail and comprehensively the roles of apoptosis and autophagy in the development of CRC. It is a well-written article, although before publication I would recommend minor changes to its content and structure.
1. First, a review article should not only be a simple revision of the findings in that particular field of research but also has to evaluate and analyze the current knowledge, in the present case of apoptosis/autophagy in CRC. Based on this the reader should get new insights, and new associations of different standpoints, that would ground further analytical studies. Briefly, I suggest to the Authors to include some of their own ideas, viewpoint regards the current state of the art regarding this topic.
2. The conclusion section seems to be a continuation of the discussion. The information between lines 428-468 should be moved to the appropriate sections of the manuscript and in this section only a short summary of the present article and the opinion of the Authors (as an added value to the current knowledge) should be retained.
Kind regards
Reviewer 2 Report
Orlandi et al. outlined the various functions of autophagy and apoptosis in the growth and development of colorectal cancer. Although the subject is incredibly intriguing, there are a number of issues that need to be addressed.
1. There are minor corrections in lines 13, 35, 55, 70, 137, 228, 231, 341, and 396.
2. The reference number in line 232 should be added.
3. All the abbreviations should be carefully revised throughout the entire manuscript.
4. The authors focused only on the roles of p53, SQSTM1/p62, and NF-κB in autophagy and apoptosis, I believe the authors should address the molecular mechanisms and signaling pathways involved in autophagy and apoptosis in CRC.
5. It is important to discuss the therapeutic strategies used to prevent autophagy and apoptosis in CRC.
6. The conclusion section should be reduced.
Reviewer 3 Report
ijms-2427356
Title: Different roles of apoptosis and autophagy in the development of human colorectal cancer
Authors: Giulia Orlandi, Luca Roncucci, Gianluca Carnevale, Paola Sena *
This review paper focuses on the different roles of apoptosis and autophagy in colorectal cancer development. Overall, the paper is well-written, but several areas require correction and revision.
[Major concerns]
1. Abbreviations: The use of abbreviations when writing a paper has many advantages besides simplicity of expression. To use an abbreviation, first write the abbreviation in parentheses after the full name, and then use the abbreviation from Introduction to the final Conclusion. Only in Abstract and Figure legend do it separately.
And since so many abbreviations are used in this article, please reorganize them in full name (abbreviation) form very thoroughly from the introduction.
2. English: There are many typos. There are also many cases where the first letter of a word is capitalized even though it is not a proper noun.
3. Although the content is well-described, I have identified a few instances where the same protein and cancer cell names are expressed in two different forms. I recommend ensuring consistency by using a single form throughout the paper. Examples: BCL-2 vs. Bcl-2; HT29 cells at Lines 265 and 403 vs. HT-29 cells at Lines 330 and 465, etc.
4. It would be beneficial to provide the full names of the abbreviations used in each figure in the respective figure legends.
[Minor concerns]
1. Line 55: Signal should be written as signal.
2. Figure 3: NAD+ should be written as NAD+.
3. Line 180: 3-Methyladenine should be written as 3-methyladenine.
4. Line 281: NFB should be written as NF-κB.
5. Figure 4: Procaspase 9 should be written as Pro-caspase 9 as other pro-caspases.
6. Reference section: Author should consult and peruse carefully recent issues of the journal, International Journal of Molecular Sciences (IJMS), for format and style. Also double-check the abbreviations of journal names. There are even many missing pages in several references.
Overall, the manuscript can be considered to publication after major revision as indicated above.
ijms-2427356
Title: Different roles of apoptosis and autophagy in the development of human colorectal cancer
Authors: Giulia Orlandi, Luca Roncucci, Gianluca Carnevale, Paola Sena *
This review paper focuses on the different roles of apoptosis and autophagy in colorectal cancer development. Overall, the paper is well-written, but several areas require correction and revision.
[Major concerns]
1. Abbreviations: The use of abbreviations when writing a paper has many advantages besides simplicity of expression. To use an abbreviation, first write the abbreviation in parentheses after the full name, and then use the abbreviation from Introduction to the final Conclusion. Only in Abstract and Figure legend do it separately.
And since so many abbreviations are used in this article, please reorganize them in full name (abbreviation) form very thoroughly from the introduction.
2. English: There are many typos. There are also many cases where the first letter of a word is capitalized even though it is not a proper noun.
3. Although the content is well-described, I have identified a few instances where the same protein and cancer cell names are expressed in two different forms. I recommend ensuring consistency by using a single form throughout the paper. Examples: BCL-2 vs. Bcl-2; HT29 cells at Lines 265 and 403 vs. HT-29 cells at Lines 330 and 465, etc.
4. It would be beneficial to provide the full names of the abbreviations used in each figure in the respective figure legends.
[Minor concerns]
1. Line 55: Signal should be written as signal.
2. Figure 3: NAD+ should be written as NAD+.
3. Line 180: 3-Methyladenine should be written as 3-methyladenine.
4. Line 281: NFB should be written as NF-κB.
5. Figure 4: Procaspase 9 should be written as Pro-caspase 9 as other pro-caspases.
6. Reference section: Author should consult and peruse carefully recent issues of the journal, International Journal of Molecular Sciences (IJMS), for format and style. Also double-check the abbreviations of journal names. There are even many missing pages in several references.
Overall, the manuscript can be considered to publication after major revision as indicated above.
Round 2
Reviewer 2 Report
The authors covered all my concerns, and I recommend accepting the review.
Author Response
We acknowledge the extensive work of the reviewer that has allowed a significant improvement of the quality of the manuscript.
Reviewer 3 Report
ijms-2427356-v2
Title: Different roles of apoptosis and autophagy in the development of human colorectal cancer
Authors: Giulia Orlandi, Luca Roncucci, Gianluca Carnevale, Paola Sena *
The revised manuscript has been greatly improved and has been very helpful for reading and understanding. However, following issues need to be considered prior to considering the manuscript of publication.
[Major concerns]
English: I mentioned it at the 1st review, but I am extremely perplexed that there have been no corrections made. However, I will point it out once again. Overall, the English writing is well done, but some compound or protein names are written in uppercase letters even though they are not the first letter of the sentence or proper nouns. General compounds should be written in lowercase throughout the text.
[Minor concerns]
1. Figure 2: The notation for BCL-2 in Figure 2 should be consistent with the notation used in the main text.
2. Figure legends: When listing the full names of the abbreviations used in the figures, please follow the following format and correct them all. BAX, Bcl-2 associated X protein; Bcl-2, B-cell lymphoma-2; mTOR, mammalian target of rapamycin; NF-κB, nuclear factor-κB; PARP, poly(ADP-ribose) polymerase, SQSTM1/p62, sequestosome 1. Please do not put ‘abbreviations’ here. When listing the abbreviations, it would be helpful for readers to easily locate and understand them if they are arranged in alphabetical order.
3. Lines 68 and more: "In the previous page at Line 32, it was stated that colorectal cancer would be abbreviated as CRC. However, there are subsequent sentences where it is repeatedly written as 'colorectal cancer.' Therefore, if there is no reason to use the abbreviation CRC, please find and correct all instances. There are similar cases like this, so please make the necessary corrections throughout.
4. Lines 217 and 221, and more: SQSTM1/p62 vs. p62/SQSTM1. In addition to this, please ensure to properly verify and correct the accurate notation of other terms."
5. Line 504: ‘Casein kinase 1 α’ should be written as ‘casein kinase 1α’.
6. Reference section: Author should consult and peruse carefully recent issues of the journal, International Journal of Molecular Sciences (IJMS), for format and style. Also double-check the abbreviations of journal names.
Overall, the manuscript can be considered to publication after major revision as indicated above.
ijms-2427356-v2
Title: Different roles of apoptosis and autophagy in the development of human colorectal cancer
Authors: Giulia Orlandi, Luca Roncucci, Gianluca Carnevale, Paola Sena *
The revised manuscript has been greatly improved and has been very helpful for reading and understanding. However, following issues need to be considered prior to considering the manuscript of publication.
[Major concerns]
English: I mentioned it at the 1st review, but I am extremely perplexed that there have been no corrections made. However, I will point it out once again. Overall, the English writing is well done, but some compound or protein names are written in uppercase letters even though they are not the first letter of the sentence or proper nouns. General compounds should be written in lowercase throughout the text.
[Minor concerns]
1. Figure 2: The notation for BCL-2 in Figure 2 should be consistent with the notation used in the main text.
2. Figure legends: When listing the full names of the abbreviations used in the figures, please follow the following format and correct them all. BAX, Bcl-2 associated X protein; Bcl-2, B-cell lymphoma-2; mTOR, mammalian target of rapamycin; NF-κB, nuclear factor-κB; PARP, poly(ADP-ribose) polymerase, SQSTM1/p62, sequestosome 1. Please do not put ‘abbreviations’ here. When listing the abbreviations, it would be helpful for readers to easily locate and understand them if they are arranged in alphabetical order.
3. Lines 68 and more: "In the previous page at Line 32, it was stated that colorectal cancer would be abbreviated as CRC. However, there are subsequent sentences where it is repeatedly written as 'colorectal cancer.' Therefore, if there is no reason to use the abbreviation CRC, please find and correct all instances. There are similar cases like this, so please make the necessary corrections throughout.
4. Lines 217 and 221, and more: SQSTM1/p62 vs. p62/SQSTM1. In addition to this, please ensure to properly verify and correct the accurate notation of other terms."
5. Line 504: ‘Casein kinase 1 α’ should be written as ‘casein kinase 1α’.
6. Reference section: Author should consult and peruse carefully recent issues of the journal, International Journal of Molecular Sciences (IJMS), for format and style. Also double-check the abbreviations of journal names.
Overall, the manuscript can be considered to publication after major revision as indicated above.
